# Impact of Lymphovascular Invasion on Prognosis in the Patients with Bladder Cancer—Comparison of Transurethral Resection and Radical Cystectomy

**DOI:** 10.3390/diagnostics11020244

**Published:** 2021-02-04

**Authors:** Kei Yoneda, Naoto Kamiya, Takanobu Utsumi, Ken Wakai, Ryo Oka, Takumi Endo, Masashi Yano, Nobuyuki Hiruta, Tomohiko Ichikawa, Hiroyoshi Suzuki

**Affiliations:** 1Department of Urology, Toho University Sakura Medical Center, 564-1 Shimoshizu, Sakura-shi, Chiba 285-8741, Japan; kei.y612@gmail.com (K.Y.); takanobu.utsumi@med.toho-u.ac.jp (T.U.); ryou.oka@med.toho-u.ac.jp (R.O.); takumi.endou@med.toho-u.ac.jp (T.E.); masashi.yano@med.toho-u.ac.jp (M.Y.); hiroyoshi.suzuki@med.toho-u.ac.jp (H.S.); 2Department of Urology, Chiba University Graduate School of Medicine, 1-8-1 Inohana, Chuo-ku, Chiba-city, Chiba 260-8687, Japan; promisedland87@gmail.com (K.W.); tomohiko_ichikawa@faculty.chiba-u.jp (T.I.); 3Department of Surgical Pathology, Toho University Sakura Medical Center, 564-1 Shimoshizu, Sakura-shi, Chiba 285-8741, Japan; nhr@med.toho-u.ac.jp

**Keywords:** bladder cancer, lymphovascular invasion, transurethral resection of bladder tumor, radical cystectomy, neoadjuvant chemotherapy

## Abstract

(1) Background: This study aimed to evaluate the associations of lymphovascular invasion (LVI) at first transurethral resection of bladder (TURBT) and radical cystectomy (RC) with survival outcomes, and to evaluate the concordance between LVI at first TURBT and RC. (2) Methods: We analyzed 216 patients who underwent first TURBT and 64 patients who underwent RC at Toho University Sakura Medical Center. (3) Results: LVI was identified in 22.7% of patients who underwent first TURBT, and in 32.8% of patients who underwent RC. Univariate analysis identified ≥cT3, metastasis and LVI at first TURBT as factors significantly associated with overall survival (OS) and cancer-specific survival (CSS). Multivariate analysis identified metastasis (hazard ratio (HR) 6.560, *p* = 0.009) and LVI at first TURBT (HR 9.205, *p* = 0.003) as significant predictors of CSS. On the other hand, in patients who underwent RC, ≥pT3, presence of G3 and LVI was significantly associated with OS and CSS in univariate analysis. Multivariate analysis identified inclusion of G3 as a significant predictor of OS and CSS. The concordance rate between LVI at first TURBT and RC was 48.0%. Patients with positive results for LVI at first TURBT and RC displayed poorer prognosis than other patients (*p* < 0.05). (4) Conclusions: We found that the combination of LVI at first TURBT and RC was likely to provide a more significant prognostic factor.

## 1. Introduction

The presence of lymphovascular invasion (LVI) has been suggested to predict poor prognosis of bladder cancer, such as more advanced disease and recurrence, and has been reported as a poor prognostic factor even for other carcinomas [1,2,3,4,5,6,7,8,9,10]. As transurethral resection of bladder tumor (TURBT) is a standard treatment for ≥cT1 bladder cancer, the utility of identifying LVI in specimens of TURBT has been suggested [1,2,3,4,5,6,7,9,10,11,12,13]. The presence of LVI in TURBT specimens has been associated with pathologic upstaging at radical cystectomy and reduced recurrence-free survival (RFS) and progression-free survival [11]. Radical cystectomy (RC) is a standard treatment for muscle-invasive bladder cancer (MIBC), and the presence of LVI in RC specimens has also been associated with poor prognosis [14,15,16,17].

Further, LVI has been described as a risk factor in the guidelines of the American Urological Association. Those guidelines define high-grade and T1 tumors, recurrent high-grade and Ta tumors, high-grade Ta and large (>3 cm) tumor, multifocal high-grade Ta tumor, any carcinoma in situ (CIS), any Bacille de Calmette et Guérin failure in high-grade cases, any variant history, any high-grade prostatic urethral involvement, and any LVI as high-risk factors [18]. However, other guidelines such as those of the European Association of Urology do not define LVI as a high-risk factor [19]. Although T stage and grade are common high-risk factors, LVI has also been reported as independently associated with recurrence and progression, even when limited to pT1 high-grade patients [20]. We thus consider that LVI is also likely to represent an important high-risk factor in patients with other high-risk factors, such as T1 and high-grade bladder cancers. High-risk factors are important because the frequency of progression from non-MIBC (NMIBC) to MIBC is higher for high-risk patients than for low-risk patients. NMIBC progresses to MIBC in about 15% of patients, resulting in poor prognosis [21]. Although the standard treatment for MIBC is RC, up to 50% of patients treated in this manner experience disease recurrence and mortality [22]. For NMIBC patients, assessment of LVI at TURBT is very important for risk stratification and decision-making regarding further treatment. In comparison, for patients after RC, assessment of LVI at RC is suggested to help in determining adjuvant therapy.

As noted above, LVI has been studied in case reports of small numbers of cases, in larger international collaborative groups, and in meta-analyses. In particular, an extensive body of literature exists regarding LVI at RC, which is strongly suggested to be associated with poor clinical outcomes. On the other hand, LVI at TURBT has also been increasingly reported recently, and is suggested to be associated with poor prognosis. TURBT is the first-line treatment for bladder cancer, and we considered first-time (first) TURBT as the most important for diagnosis and treatment. Although many investigations have examined TURBT and RC individually, few have looked into LVI at TURBT and RC.

The present study aimed to evaluate the relationship between LVI on both TURBT and RC, as well as overall survival (OS) and cancer-specific survival (CSS). The present study also aimed to assess the concordance rate between TURBT and RC specimens with regard to the presence of LVI. Furthermore, we analyzed differences in prognosis due to the combination of LVI at TURBT and RC.

## 2. Materials and Methods

This was a retrospective study approved by the institutional review board (approval number: S17108, approval date. 17 July 2017). We identified 423 patients with bladder cancer treated by TURBT at Toho University Sakura Medical Center between January 2012 and December 2016. Moreover, a total of 291 patients underwent first TURBT for bladder cancer. Of those, 75 patients were excluded from the present study due to the following reasons: tumor of any other origin (*n* = 1); concurrent or recurrent bladder cancer with upper urinary tract urothelial carcinoma (UC) (*n* = 16); bladder Tis lesions only (*n* = 8); and missing values (*n* = 50). A final total of 216 patients were analyzed for prognostic factors of first TURBT, and to determine the utility of LVI (Appendix A).

Furthermore, we analyzed prognostic factors for RC, and considered the relationship between LVI in samples from RC and LVI in samples from first TURBT. A total of 75 patients who underwent RC for bladder cancer were seen at our institution between 2010 and 2017. Of those, 11 patients were excluded for the following reasons: the pathology was not UC (*n* = 2); and missing values (*n* = 9). Eventually, a total of 64 patients were analyzed regarding prognostic factors for RC (Appendix A). Twenty patients (31.3%) received neoadjuvant chemotherapy (NAC). Generally, we administered two cycles of gemcitabine-cisplatin (GC). In addition, 50 patients underwent both first TURBT and RC at our institution, and we analyzed the concordance rate of each LVI. More details can be seen in the flow chart in Figure 1.

Medical records were surveyed retrospectively, and the following data were collected from medical charts: age at surgery, sex, smoking status, tumor size, Multiple tumors, Appearance, clinical T stage, intravesical instillation, presence or absence of NAC, pathological T stage, grade (G1, G2, and G3), presence of metastasis, urinary cytology, histological variant, CIS, and LVI. LVI was defined as the presence of tumor cells within an arterial, venous, or lymphatic lumen. Presence of LVI in specimens was assessed using conventional hematoxylin and eosin (HE) staining. If we encountered any difficulty in evaluations using only HE staining, we added Elastica van Gieson staining. Positive urinary cytology was defined as ≥class IV (malignancy suspected) according to Papanicolaou’s classification. Lymph node metastasis and distant metastasis were collectively defined as metastasis. Pathological evaluation was performed by two out of three pathologists each as a council system in Toho University Sakura Medical Center using the tumor, node, and metastasis classification system updated in 2009 and the new classification for grading non-invasive urothelial bladder carcinomas proposed by the World Health Organization and the International Society of Urological Pathology in 2004 [7,8,9,10].

### Statistical Analysis

Results are presented as median and range or mean ± standard deviation, as appropriate. Continuous parametric variables were compared using *t*-tests. Non-parametric variables were compared using Mann-Whitney *U* tests. The Kaplan-Meier method was used to estimate OS and CSS following TURBT and RC, respectively. The log-rank test was used to compare statistical significances in each curve. After variables were selected in univariate analyses, regression analyses of Cox proportional hazards were performed to determine factors significantly associated with OS and CSS.

These statistical analyses were carried out using JMP Pro version 13 (SAS Institute, Cary, NC, USA). All statistical tests were two-sided, and values of *p* < 0.05 were considered significant.

## 3. Results

### 3.1. LVI at First TURBT

The clinical and pathological characteristics of patients who underwent first TURBT are shown in Appendix A (*n* = 216). Mean patient age was 71.3 ± 9.3 years, and males represented 79.6% of the study population. G3 was present in 43.5% of patients, 22.7% of patients had LVI, and only 9.7% had CIS. We compared the LVI-positive and -negative groups (Table 1). The LVI-positive group showed higher clinical stage (*p* < 0.001), pathological T stage (*p* < 0.001), non-papillary (*p* < 0.001), G3 (*p* < 0.001), and histological variant (*p* < 0.001), compared to the LVI-negative group. Subsequently, we analyzed prognostic factors from first TURBT (Table 2). Univariate analyses identified cT stage, metastasis, histological variant, BCG intravesical instillation after TURBT and LVI as significantly associated with OS (*p* < 0.001, *p* < 0.001, *p* < 0.001, *p* = 0.044, *p* = 0.003, respectively). Multivariate analysis identified metastasis (hazard ratio (HR) 11.363, 95% confidence interval (CI) 3.968–32.285; *p* <0.001) as significant predictors of OS after first TURBT. According to univariate analyses, ≥cT3 (*p* < 0.001), metastasis (*p* < 0.001), histological variant (*p* < 0.001), and LVI (*p* < 0.001) were significantly associated with CSS (Appendix A). Patients in the LVI-negative group at first TURBT (*n* = 167) displayed significantly better CSS than those in the LVI-positive group at first TURBT (*n* = 49; *p* < 0.001). Multivariate analysis identified metastasis (HR 13.514, 95%CI 3.125–58.824; *p* < 0.001) and LVI (HR 7.407, 95%CI 1.328–41.667; *p* = 0.022) as significant predictors of CSS.

### 3.2. LVI at RC

Clinical and pathological characteristics of patients who underwent RC are shown in Appendix A (*n* = 64). Mean patient age was 68.2 ± 7.9 years, and males represented 79.7% of the study population. G3 was present in 53.1% of patients, 32.8% of patients had LVI, and 10.9% had CIS. Moreover, in 13 patients with lymph node metastasis, 9 patients had positive LVI at RC (*p* = 0.046). We analyzed the survival implications in 64 patients who underwent RC (Table 3). In univariate analyses, pT stage, tumor grade and LVI were significantly associated with OS (*p* < 0.001, *p* = 0.001, *p* = 0.001, respectively). Multivariate analysis identified presence of G3 (HR 7.942, 95%CI 1.343–150.803; *p* = 0.020) as a significant predictor of OS. According to univariate analyses, ≥cT3, presence of G3 and LVI were significantly associated with CSS (*p* = 0.002, *p* = 0.003, *p* = 0.001, respectively) (Appendix A). Patients in the LVI-negative group at RC (*n* = 43) exhibited significantly better CSS than those in the LVI-positive group at RC (*n* = 21; *p* = 0.001). Multivariate analysis identified presence of G3 (HR 6.373, 95%CI 1.012–123.091; *p* = 0.048) as the only significant predictor of CSS.

### 3.3. Comparison of LVI between First TURBT and RC

We analyzed the concordance rate between LVI at first TURBT and LVI at RC (Table 3). We analyzed data from the 50 patients who underwent RC, excluding 14 patients who did not undergo first TURBT in our hospital or for whom data were missing. Of the 31 patients with LVI at first TURBT, 10 patients (32.3%) also showed LVI at RC. Of the 19 patients without LVI at first TURBT, 14 patients (73.7%) were also found to have no LVI at RC (Table 4).

We confirmed the effect of NAC on LVI in RC specimens. The clinical and pathological characteristics of patients according to use of NAC are shown in Appendix A. In the group receiving NAC (NAC+ group) at first TURBT, cT stage was higher (*p* = 0.006), and LVI was more frequently evident (*p* = 0.003) compared to the group not receiving NAC (NAC- group). On the other hand, patients in the NAC+ group at RC displayed significantly higher pT stage than those in the NAC- group (*p* = 0.044). No significant difference in the presence or absence of LVI was evident between NAC+ and NAC− groups at RC (*p* = 0.312). In patients showing LVI (LVI+) at first TURBT, but not at RC, pT stage was pT0 in 9 patients (42.9%). Appendix A shows changes in LVI according to the use of NAC in LVI+ cases at first TURBT. LVI in RC specimens showed no significant difference according to use of NAC. No significant difference in LVI+ or LVI- status was seen between the NAC+ group (*n* = 14) and NAC− group (*n* = 17) in LVI+ cases at first TURBT (*p* = 0.252).

Finally, survival analysis was conducted across all combinations of LVI at first TURBT and RC. Patients with no evidence of LVI at both first TURBT and RC (TURBT−/RC−) had the best prognosis, and patients with evidence of LVI at both first TURBT and RC (TURBT+/RC+) had the worst prognosis (*p* = 0.016) (Figure 2). Five-year CSS rates for each combination of LVI at first TURBT and RC were determined. Five-year CSS rate was 92.9% for patients with TURBT−/RC−, 80.0% for those with LVI at RC but not first TURBT (TURBT−/RC+), 77.0% for those with LVI at first TURBT but not at RC (TURBT+/RC−), and 50.0% for those with evidence of LVI at both first TURBT and RC (TURBT+/RC+).

## 4. Discussion

This study detected LVI in 22.7% of first TURBT specimens and in 32.8% of RC specimens. These rates were comparable to those described in the literature. The rate of LVI at TURBT has been reported as 6%–70%, and the rate of LVI at RC has been reported to range from 30% to 50% [1]. Clinical stage and pathological T stage were higher in the LVI-positive group than in the LVI-negative group at first TURBT. Moreover, we found that patients with LVI often showed presence of G3 in the resected specimen. Patients with LVI were thus suggested to be more likely to be at high risk. In clinical practice, choosing the next treatment for patients with high-grade pT1 is difficult. This is because high-risk bladder cancer is more likely to progress to MIBC, and RC is also a treatment option. Mathieu et al. showed that high-grade pT1 bladder cancer and LVI detected in biopsy specimens were poor prognostic factors for disease recurrence and progression [23]. RC is thus suggested to be a treatment option before progression for high-grade T1 patients with LVI.

At first TURBT, ≥cT3, presence of metastasis, histological variant, and LVI were found to be significantly associated with OS and CSS in univariate analyses. However, in the patients with only lymph node metastasis, univariate analysis showed a significant difference in OS, but multivariate analysis did not about OS (*p* < 0.001, *p* = 0.47). On the other hand, LVI was found to be a significant predictor of CSS in multivariate analysis. Various studies have reported that LVI was associated with CSS, but not with OS. A meta-analysis of LVI at first TURBT has shown that the pooled hazard ratio was significant for CSS (HR 1.35, 95%CI 1.01–1.81; *p* = 0.04), but not for OS (HR 1.55, 95%CI 0.90–2.67; *p* = 0.11) [11]. Moreover, Streeper et al. showed that patients with LVI of clinical stage I or II showed lower survival than those without LVI (HR 2.68, 95%CI 1.55–4.64; *p* = 0.049) [3]. Resnick et al. also associated LVI at TURBT with an increasing likelihood of node-positive disease (48.3% vs. 25.0%; *p* < 0.001) [24]. In addition, the association between LVI in TURBT specimens and RFS has also been reported, and LVI at TURBT is a poor prognostic factor [6,20,25].

Subsequently, at RC, factors of ≥pT3, presence of G3 and presence of LVI were found to be significantly associated with OS and CSS in univariate analyses. In multivariate analysis, inclusion of G3 was identified as a significant predictor of OS and CSS. High tumor grade is included as a high-risk factor in all guidelines, and it is also included as a factor in the nomogram that predicts risk of recurrence after RC [26]. On the other hand, LVI was shown to be significantly associated with OS and CSS in univariate analysis. Although LVI was not identified as a significant factor in multivariate analysis in the present study, several studies have reported LVI as associated with OS, CSS, and RFS even in multivariate analysis [16,27,28,29,30,31,32]. Mari et al. summarized the relationship between LVI and clinical outcomes after RC [15]. They found that patients with LVI at RC exhibited higher risks of disease recurrence (HR 1.57, 95%CI 1.45–1.70) and cancer-specific mortality (HR 1.59, 95%CI 1.48–1.73). Furthermore, they found that LVI was associated with recurrence and cancer-specific mortality with lymph-node-negative bladder cancer particularly. In this study, five patients who underwent RC had lymph node metastasis, and positive LVI was seen at TURBT in all these patients. Five out of seven patients who were diagnosed no lymph node metastasis before Surgery but positive LVI at first TURBT before Surgery had lymph node metastasis at RC. We suggest that lymph node-negative patients with LVI were more likely to have lymph node metastasis in the future, and LVI was likely to be a risk factor for recurrence and cancer-specific mortality.

Moreover, we considered the relationship between NAC and LVI. Although there was no report that evaluated LVI before and after total resection as in this study, some reports discussed that the presence or absence of LVI after NAC became a prognostic factor in bladder cancer and other cancers [33,34,35]. In this study, there was no significant difference in the disappearance of LVI at RC according to the use of NAC in the LVI+ group at first TURBT (*p* = 0.282). However, in 14 patients who were diagnosed with LVI + at TURBT and underwent NAC, the RC LVI negative group had a significantly better prognosis than the RC LVI positive group (*p* = 0.03867) (Figure 3). Although the number of cases is small, it was suggested that the disappearance of LVI at RC may be a predictor of the effect of NAC and prognosis of bladder cancer.

Next, we considered the concordance between LVI in first TURBT and RC specimens. A total of 10 patients (32.2%) with LVI at first TURBT showed the presence of LVI at RC. On the other hand, a total of 14 patients (73.7%) without LVI at first TURBT had negative LVI at RC. In other words, the concordance rate between LVI at TURBT and at RC was 48.0%. The concordance rate was about 60%–70% in some other reports, but was lower in this study [3,24,36]. Moreover, we analyzed differences in prognosis due to the combination of LVI at first TURBT and RC, and found an interesting result: all patients with evidence of LVI at both first TURBT and RC displayed very poor prognosis and died within 2 years. In this study, LVI does not show a very high concordance rate. However, when both LVI at TURBT and RC are positive, it may offer very useful information. Consequently, in patients with LVI at both first TURBT and RC, the results may be useful for considering adjuvant therapy after RC, such as adjuvant chemotherapy, to improve prognosis.

## 5. Conclusions

LVI at first TURBT and RC were associated with OS and CSS. Specifically, LVI at first TURBT was suggested to represent an independent prognostic factor for CSS. Furthermore, we found that the combination of LVI at first TURBT and RC was likely to offer a more significant prognostic factor than either factor individually.

## Figures and Tables

**Figure 1 diagnostics-11-00244-f001:**
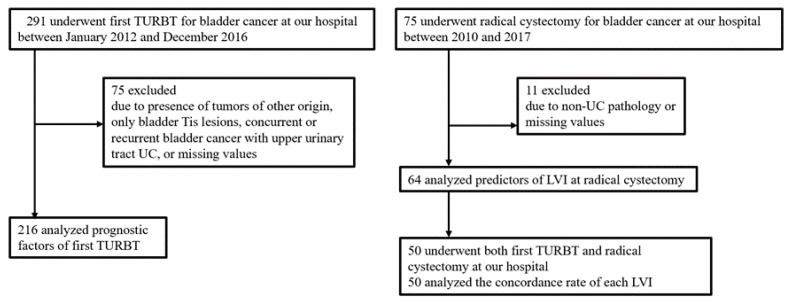
Flow chart for patient selection. TURBT: transurethral resection of bladder tumor; UC: urothelial carcinoma; LVI: lymphovascular invasion.

**Figure 2 diagnostics-11-00244-f002:**
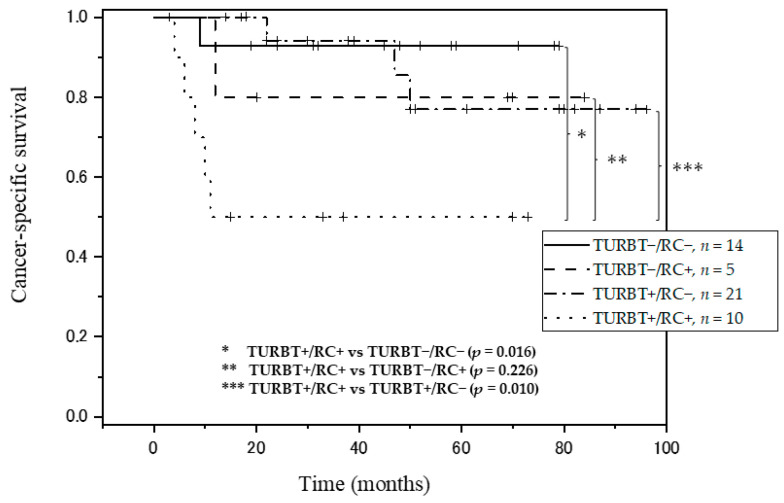
Kaplan-Meier curves for cancer-specific survival in all combinations of LVI at first TURBT and RC. Cancer-specific survival in * TURBT+/RC+ vs TURBT−/RC− (*p* = 0.016), ** TURBT+/RC+ vs TURBT−/RC+ (*p* = 0.226), *** TURBT+/RC+ vs TURBT+RC− (*p* = 0.010). LVI: lymphovascular invasion; TURBT: transurethral resection of bladder tumor; RC: radical cystectomy.

**Figure 3 diagnostics-11-00244-f003:**
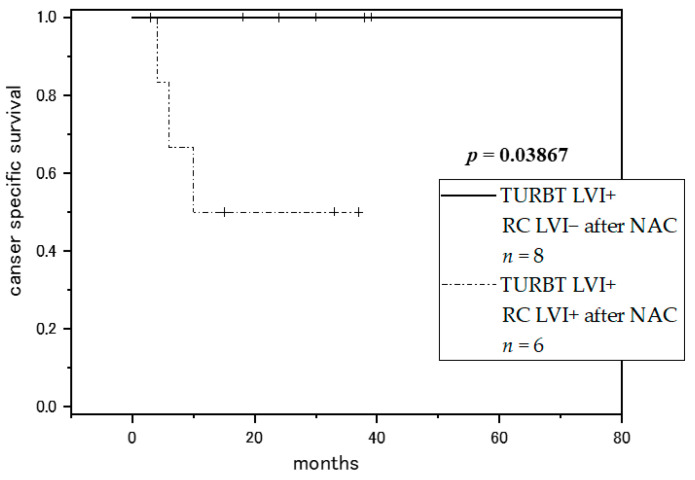
Kaplan-Meier curves for cancer-specific survival in the patients who were diagnosed with LVI + at TURBT and underwent NAC, RC LVI− vs. RC LVI+. LVI: lymphovascular invasion; TURBT: transurethral resection of bladder tumor; RC: radical cystectomy.

**Table 1 diagnostics-11-00244-t001:** Clinical and pathological characteristics stratified by LVI at first TURBT.

		LVI+ *n* = 49	LVI− *n* = 167	*p*-Value
Age, years (range)		74.0 (43–89)	71.0 (40–92)	0.30
Males, *n* (%)		38 (78.6)	134 (80.2)	0.73
Smoking status, n (%)		32 (65.3)	97 (58.1)	0.48
Clinical stage, *n* (%)	stage 0a, I	12 (24.5)	151 (90.4)	<0.001
	stage II	19 (38.8)	10 (6.0)	
	stage III	8 (16.3)	2 (1.2)	
	stage IV	10 (20.4)	4 (2.4)	
Tumor size (cm)		2.8 (0.1–7)	1.7 (0.1–8.5)	<0.001
Multiple tumors, n (%)		23 (46.9)	75 (44.9)	0.80
Appearance, n (%)	Non papillary	19 (38.8)	7 (4.2)	<0.001
Pathological T stage, *n* (%)	pTa	2 (4.0)	98 (58.7)	<0.001
	pT1	15 (30.0)	61 (36.5)	
	≥pT2	32 (64.0)	8 (4.8)	
presence of G3, *n* (%)		38 (75.5)	56 (33.5)	<0.001
CIS, *n* (%)		6 (12.2)	15 (9.0)	0.53
histological variant, n (%)		14 (28.6)	8 (4.8)	<0.001

LVI: lymphovascular invasion; TURBT: transurethral resection of bladder tumor; CIS: carcinoma in situ.

**Table 2 diagnostics-11-00244-t002:** Uni- and multivariate analyses of clinicopathological variables for OS and CSS after first TURBT.

	OS		CSS	
	Univariate *p*-Value	Multivariate HR (95%CI) *p*-Value	Univariate *p*-Value	Multivariate HR (95%CI) *p*-Value
cT stage (<3/≥3)	<0.001	2.2553 0.244	<0.001	2.0348 0.433
Metastasis (−/+)	<0.001	11.363 3.968–32.258 <0.001	<0.001	13.514 3.125–58.824 <0.001
Grade (G1, 2/G3)	0.61		0.16	
LVI (−/+)	0.003	1.3092 0.665	<0.001	7.407 1.328–41.667 0.022
CIS (−/+)	0.23		0.36	
Urinary cytology (−/+)	0.91		0.88	
histological variant (−/+)	<0.001	1.6083 0.446	<0.001	1.5683 0.558
BCG intravesical instillation after TURBT (−/+)	0.044	0.2412 0.170	0.09	

OS: overall survival; CSS: cancer-specific survival; TURBT: transurethral resection of bladder tumor; RC: radical cystectomy; HR: hazard ratio; NAC: neoadjuvant chemotherapy; LVI: lymphovascular invasion; CIS: carcinoma in situ.

**Table 3 diagnostics-11-00244-t003:** Uni- and multivariate analyses of clinicopathological variables for OS and CSS after RC.

	OS		CSS	
	Univariate *p*-Value	Multivariate HR (95%CI) *p*-Value	Univariate *p*-Value	MultivariateHR (95%CI) *p*-Value
pT stage (<3/≥3)	<0.001	1.8240 0.5354–6.6247 0.34	0.002	1.7037 0.4602–6.7589 0.42
Metastasis (−/+)	0.49		0.75	
NAC (−/+)	0.71		0.91	
Grade (G1, 2/G3)	0.001	7.9418 1.3427–150.8031 0.020	0.003	6.3727 1.0124–123.0906 0.048
LVI (−/+)	0.001	2.2189 0.6237–8.4612 0.22	0.001	2.6153 0.6663–11.4548 0.17
CIS (−/+)	0.63		0.74	
histological variant (−/+)	0.70		0.83	
BCG intravesical instillation before RC (−/+)	0.90		0.63	

OS: overall survival; CSS: cancer-specific survival; TURBT: transurethral resection of bladder tumor; RC: radical cystectomy; HR: hazard ratio; NAC: neoadjuvant chemotherapy; LVI: lymphovascular invasion; CIS: carcinoma in situ.

**Table 4 diagnostics-11-00244-t004:** Concordance between LVI at first TURBT and RC.

	LVI+, TURBT *n* (%)	LVI−, TURBT *n* (%)	Total
LVI+, RC	10 (20%)	5 (10%)	15
LVI−, RC	21 (42%)	14 (28%)	35
Total	31	19	

LVI: lymphovascular invasion; TURBT: transurethral resection of bladder tumor; RC: radical cystectomy.

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
