# Peer review of "Impact of Lymphovascular Invasion on Prognosis in the Patients with Bladder Cancer—Comparison of Transurethral Resection and Radical Cystectomy"

_diagnostics, 2021, doi:10.3390/diagnostics11020244_

Round 1
Reviewer 1 Report
In this study, the authors want evaluate the associations of lymphovascular invasion (LVI) at first transurethral resection of bladder (TURBT) and radical cystectomy (RC) with survival outcomes, and to evaluate the concordance between LVI at first TURBT and RC.
This is an important issue in bladder cancer. They concluded that combination of LVI at first TURBT and RC was likely to provide a more significant prognostic factor. Some comments are as follow.
1.Some important clinical or tumor features should be included, such as tumor size, histological variant, tumor appearance (papillary or sessile), tumor number, intravesical instillation, and smoking status. Please include these parameters in the survival analysis (Table 1 to Table 3).
2.Tumor grade should be classified as low or high. The authors used High/Low grade system in the recent published paper (Curr Urol. 2020 Oct;14(3):135-141) but used G1/G2/G3 in this study. Why?
3.In Table 1, there are 176 patines with pTa-pT. What is the intravesical instillation protocol (BCG or non-BCG)? Is LVI associated with bladder tumor recurrence in the BCG/nonBCG patients?
4.The format of Table 1 and Table 2 is not easy to read. Please modify them.
5.In the Table 2, cT stage is NOT associated with OS and CSS. Why?
6.In the Table 3, pT stage is NOT associated with OS and CSS. Why?
7.The authors concluded that LVI at first TURBT and RC was likely to provide a more significant prognostic factor. But LVI is NOT a significant predictor of OS (p=0.22) and CSS (p=0.17) in RC (Table 3) in the multivariate analysis.
8.In the Table 4, There 21 patients with LVI+ (TURBT) but LVI- (RC). Why?
In the supp Table 4, in non-NAC group, 17 patients have LVI+ in the first
TURBT but 13 of them have LVI- in RC. Why?
Author Response
Thank you for estimating our manuscript and valuable indication. We will try to reply it all the points of indications.
Answer for questions
- Some important clinical or tumor features should be included, such as tumor size, histological variant, tumor appearance (papillary or sessile), tumor number, intravesical instillation, and smoking status. Please include these parameters in the survival analysis (Table 1 to Table 3). 
  Thank you for your suggestion. As your suggestion, the above clinical parameters have been added to table 1-3. Thesefactors (Smoking status, Tumor size, Multiple tumors, histological variant) had been added to table 1 to show the association between LVI and these factors. We added the pathological findings variant and presence of BCG before and after surgery as the factors to table 2 and 3. In table2, BCG after TURBT was a factor that extended OS in univariate analysis.
- Tumor grade should be classified as low or high. The authors used High/Low grade system in the recent published paper (Curr Urol. 2020 Oct;14(3):135-141) but used  G1/G2/G3 in this study. Why?
  Thank you for your comments. As you pointed out the problem, we used the low / high classification in the paper previously submitted. However, another study suggested that G3 was a stronger factor, so we divided the cases into two groups, presence of G3 and the other in this study.
- In Table 1, there are 176 patines with pTa-pT. What is the intravesical instillation protocol (BCG or non-BCG)? Is LVI associated with bladder tumor recurrence in the BCG/nonBCG patients?
  Thank you for your comments. In principle, BCG intravesical instillation therapy is performed for patients with medium or high risk pTia-1 bladder cancer. Low-risk groups (satisfying all â‘ -â‘¥) do not undergo BCG intravesical instillation therapy.â‘ Initial, â‘¡ Single, â‘¢ Ta, â‘£ G1 (low grade), ⑤ ≦3 cm , â‘¥ absence of CIS. On the other hand, for the cases corresponded any of the following (1) – (6), they will undergo radical cystectomy without BCG intravesical instillation therapy. (1) T1 high grade tumor with extensive CIS, (2) T1 high grade tumor with CIS at prostatic urinary tract, (3) multiple and / or 3 cm or more and / or recurrent T1 high grade tumor, (4) T1 high grade tumors with variant histology such as micropapillary, (5)T1 high grade tumors with LVI, (6) BCG unresponsive tumors.
  We added analysis about BCG. Eleven out of 176 patients who underwent TURBT underwent immediate radical cystectomy without BCG intravesical instillation therapy. Moreover, 47 patients underwent BCG after TURBT and 11 patients had recurrence. One out of 11 had positive LVI and 10 patients had negative LVI (p=0.42). On the other hand, 118 out of 34 patients without BCG had recurrence. Two out of 34 had positive LVI and 32 patients had negative LVI (p=0.34). In conclusion, LVI was not associated bladder tumor recurrence in both of BCG and non-BCG.
- The format of Table 1 and Table 2 is not easy to read. Please modify them.
  Thank you for your suggestion. As your suggestion, we modified Table1, 2 and 3.
- In the Table 2, cT stage is NOT associated with OS and CSS. Why?
  Thank you for your comments. We consider that the association was not ruled out, because cT stage was a significant factor in the univariate analysis. Although it was not an independent factor in multivariate analysis in this study, there are reports that the cT stage is a prognostic factor. Increasing the number of cases may be a significant factor in multivariate analysis.
- In the Table 3, pT stage is NOT associated with OS and CSS. Why?
  Thank you very much for your comments. Similar to Q5, we consider that the association was not ruled out, because pT stage was a significant factor in the univariate analysis. Although it was not an independent factor in multivariate analysis in this study, there are reports that the pT stage is a prognostic factor. Increasing the number of cases may be a significant factor in multivariate analysis.
- The authors concluded that LVI at first TURBT and RC was likely to provide a more significant prognostic factor. But LVI is NOT a significant predictor of OS (p=0.22) and CSS (p=0.17) in RC (Table 3) in the multivariate analysis.
  Thank you for your suggestion. As you pointed out, there is no significant difference in the multivariate analysis, but a significant difference is found in the univariate analysis. Therefore, we have shown that LVI at TURBT and RC may be more useful factors in combination rather than alone.
- In the Table 4, There 21 patients with LVI+ (TURBT) but LVI- (RC). Why?
In the supp Table 4, in non-NAC group, 17 patients have LVI+ in the first
TURBT but 13 of them have LVI- in RC. Why?
  Thank you for your comments. I think the reason for these points are as follows, First, it was possible that LVI was limited to adjacent to the tumor and was completely resected by TURBT. Second, it was possible that LVI disappeared due to neoadjuvant chemotherapy.
Reviewer 2 Report
Authors evaluate the relationship between LVI on both TURBT and RC, as well as overall survival (OS) and cancer-specific survival (CSS). The present study also aimed to assess the concordance rate between TURBT and RC specimens with regard to the presence of LVI. They also evaluated presence of LVI in NAC+ and NAC- subgroups.
Abstract: “inclusion of G3” is not clear, it means presence of G3? Same at line 213
line 40: As TURBT is a standard treatment for bladder cancer: please specify patients and stage, it’s not a standard treatment for every patient.
Line 42: pathologic upstaging at radical cystectomy
Line 99: sentence repeated
Line 104: by two of three pathologists, please explain
Line 127: metastasis as significant predictor of OS: Lymph node metastasis and distant metastasis have the same predictive value? Could you provide data about lymph node metastasis alone?
Line 174: What is the pT0 rate in patients with LVI- at first TURBT? In the NAC+ group, is there a difference in response (TRG1,2,3) in LVI+ vs. LVI- subgroups?
line 226: who were not observed no lymph node metastasis before Surgery: please rephrase
line 235: the RC-LVI negative group had a significantly better prognosis than the RC-LVI negative group: please correct
Presence of LVI at RC correlates with presence of lymph node metastasis and/or presence of distant metastasis?
Author Response
Thank you for estimating our manuscript and valuable indication. We will try to reply it all the points of indications.
1. Abstract: “inclusion of G3” is not clear, it means presence of G3? Same at line 213
Thank you for your comment. As you pointed out comment, we revised it as you pointed out. (Line27, table1, Supplementary table 1,2 and 3, Line212)
2. line 40: As TURBT is a standard treatment for bladder cancer: please specify patients and stage, it’s not a standard treatment for every patient.
Thank you for your comment. we revised it as you pointed out. (Line40)
3. Line 42: pathologic upstaging at radical cystectomy
Thank you for your comment. we revised it as you pointed out. (Line42)
4. Line 99: sentence repeated
Thank you for your comment. we deleted this sentence.
5. Line 104: by two of three pathologists, please explain
Thank you for your comments. Two out of the three pathologists diagnose for each case. we revised it as you pointed out. (Line 105)
6. Line 127: metastasis as significant predictor of OS: Lymph node metastasis and distant metastasis have the same predictive value? Could you provide data about lymph node metastasis alone?
Thank you for your suggestions. As you pointed out the problem, we have added the analysis. Nine of the 11 patients with metastasis at first TURBT had only lymph node metastasis. Univariate analysis showed a significant difference in the cases with only lymph node metastasis, but multivariate analysis did not show a significant difference in OS (p<0.001, p=0.47). it is suggested that distant metastasis may be a stronger factor, but in this study, there were only two patients with distant metastasis. Therefore, we evaluated lymph node metastasis and distant metastasis together in this study. We added this point to Discussion. (Line220-222)
7. Line 174: What is the pT0 rate in patients with LVI- at first TURBT? In the NAC+ group, is there a difference in response (TRG1,2,3) in LVI+ vs. LVI- subgroups?
Thank you for your suggestions. Six out of 19 patients with LVI- at first TURBT had pT0 at RC (31.6%). Sorry, we can’t analyze it because this study did not evaluate the effect of NAC.
8. line 226: who were not observed no lymph node metastasis before Surgery: please rephrase
Thank you for your comments. we revised it as you pointed out. (Line244-245)
9. line 235: the RC-LVI negative group had a significantly better prognosis than the RC-LVI negative group: please correct
Thank you very much for your comments. we revised it as you pointed out. (Line253-254)
10. Presence of LVI at RC correlates with presence of lymph node metastasis and/or presence of distant metastasis?
Thank you for your comments. In 13 patients with lymph node metastasis, 9 patients had positive LVI at RC (p=0.046). Therefore, we suggest that LVI is associated with lymph node metastasis at RC. On the other hand, because only one case had distant metastases, it is difficult to evaluate it. we revised it as you pointed out. (Line 157)
Round 2
Reviewer 1 Report
The authors have made good responses to my questions.